# ESCRT Machinery in HBV Life Cycle: Dual Roles in Autophagy and Membrane Dynamics for Viral Pathogenesis

**DOI:** 10.3390/cells14080603

**Published:** 2025-04-16

**Authors:** Jia Li, Reinhild Prange, Mengji Lu

**Affiliations:** 1Institute for Virology, University Hospital Essen, University of Duisburg-Essen, 45122 Essen, Germany; jiali201093@163.com; 2Institute for Virology, University Medical Center, Johannes Gutenberg University Mainz, 55131 Mainz, Germany; prange@uni-mainz.de

**Keywords:** autophagy, endosome, ESCRT, HBV

## Abstract

The endosomal sorting complexes required for transport (ESCRT) comprise a fundamental cellular machinery with remarkable versatility in membrane remodeling. It is multifunctional in the multivesicular body (MVB) biogenesis, exosome formation and secretion, virus budding, cytokinesis, plasma membrane repair, neuron pruning, and autophagy. ESCRT’s involvement in cellular mechanisms extends beyond basic membrane trafficking. By directly interacting with autophagy-related (ATG) proteins and facilitating autophagosome-lysosome fusion, ESCRT ensures cellular homeostasis. Dysregulation in ESCRT function has been implicated in cancer, neurodegenerative disorders, and infectious diseases, underscoring its critical role in numerous pathologies. Hepatitis B virus (HBV) is an enveloped virus that exploits ESCRT and autophagy pathways for viral replication, assembly, and secretion. This review synthesizes recent mechanistic insights into ESCRT’s multifaceted roles, particularly focusing on its interactions with autophagy formation and the HBV lifecycle.

## 1. Introduction

The endosomal sorting complexes required for transport (ESCRT) machinery was initially defined as a ubiquitin-dependent protein-sorting pathway. The machinery comprises four main complexes: ESCRT-0, -I, -II, and -III complex, along with the terminal AAA ATPase Vacuolar Protein Sorting-associated 4 (VPS4) [1]. Table 1 shows the subunit compositions of ESCRT machinery in mammals, yeast, and drosophila. Briefly, in mammals, ESCRT-0 is comprised of hepatocyte growth factor-regulated tyrosine kinase substrate (HRS/HGS) and signal transducing adaptor molecule (STAM1/2). ESCRT-I is a soluble hetero-tetramer consisting of Tumor Susceptibility Gene 101 protein (TSG101), VPS28, VPS37A/B/C/D, and MVB12A/B. ESCRT-II is a hetero-tetrameric protein complex, containing ELL-Associated Protein (EAP) 45, EAP30, and two EAP20 molecules. The ESCRT-III complex consists of Charged Multivesicular Body Protein (CHMP) 1A/B, CHMP2A/B, CHMP3, CHMP4A/B/C, CHMP5, CHMP6, CHMP7, and CHMP8. The VPS4 complex is formed by the type I AAA-ATPase VPS4A/B and its co-factor Vesicle Trafficking 1 (VTA1). The ESCRT machinery is crucial for MVB biogenesis, membrane protein degradation, lysosome formation, and the budding processes of several viruses [2,3].

Autophagy maintains cellular homeostasis by degrading unnecessary or damaged components. It enables the recycling of biosynthetic precursors, supports cell survival, and ensures cellular homeostasis under stress [4]. The autophagy process involves forming and expanding a pre-autophagosomal structure (PAS), encapsulating cytoplasmic material within a phagophore, sealing the phagophore into a double-membraned autophagosome, and ultimately, fusing with a lysosome for degradation [5]. In recent decades, it has become clear that the ESCRT machinery is required for autophagy in humans [6], but the mechanisms are still elusive.

Hepatitis B virus (HBV) causes acute and chronic infection in humans. HBV infection remains a significant global health challenge [7]. There are about 296 million chronically HBV-infected people worldwide. Endosomal vesicle trafficking, autophagy, and their interconnection collaboratively participate in HBV replication, assembly, and secretion [8]. ESCRT machinery-involved endosomal and autophagic pathways regulate the HBV lifecycle.

Here, we review the diverse functions of the ESCRT pathway and examine its connection to the autophagy process and lysosome functions, revealing its involvement in cancers, neurodegenerative disorders, and infectious diseases. A focus of this review is the role of ESCRT components in HBV replication and production.

## 2. ESCRT-Dependent MVB Biogenesis

The MVB is an endosomal organelle characterized by a single membrane enclosing intraluminal vesicles (ILVs). MVBs play a crucial role in cellular cargo transport by directing cargo either for lysosomal degradation or for extracellular release as exosomes. MVB biogenesis occurs at least through two mechanisms: an ESCRT-independent process involving lipid rafts, which can be disrupted by lipid metabolism inhibitors [9], and an ESCRT-dependent pathway.

The ESCRT-dependent pathway is a highly orchestrated, stepwise process involving membrane remodeling and cargo sorting. As schematized in prior studies [2,10], ESCRT complexes sequentially orchestrate key components required for MVB/exosome formation. Initially, ESCRT-0 initiates the process by identifying and concentrating ubiquitinated cargo for MVB inclusion [11,12]. ESCRT-I is then recruited, where it interacts with ESCRT-0 and ESCRT-II through the ubiquitin E2 variant (UEV) domain of TSG101 [13] and the C-terminal domain of VPS28 [14], respectively. The Y-shaped heterotetramer ESCRT-II further amplifies cargo sorting. ESCRT-I and ESCRT-II localize to membrane necks, cooperating to form cargo-containing buds through membrane evagination [11]. Apoptosis-linked gene 2 (ALG-2)-interacting protein X (Alix, AIP1) is a multifunctional protein. Its N-terminal BRO1 domain, central V-shaped domain, and C-terminal proline-rich region (PRR) allow it to interact with ESCRT components, particularly CHMP4B [1]. ESCRT-I/ESCRT-II complexes and Alix function as separate branches of the ESCRT pathway that converge on the ESCRT-III and VPS4 machinery for membrane scission [15]. The ESCRT-III complex polymerizes into spiral filaments that constrict membrane necks, while subsequent VPS4-mediated disassembly of these polymers provides the mechanochemical force required for membrane fission [16,17]. This cooperative mechanism ensures dynamic remodeling of ESCRT-III assemblies during abscission.

The ESCRT-III proteins assemble at membrane necks to cleave budding structures and form ILVs. Once the scission is complete, the VPS4 is recruited to recognize specific motifs in ESCRT-III subunits via its N-terminal microtubule interacting and trafficking (MIT) domain, leading to the dissociation of the ESCRT complex from the membrane and the recycling of its components. After the fusion of the MVB with the lysosome, ILVs and the sequestered cargo are degraded within the lysosomal lumen. Thus, the ESCRT pathway plays a crucial role in the sorting, budding, and degradation of cellular cargo.

## 3. Autophagy

While the ESCRT pathway directly facilitates membrane remodeling events (e.g., multivesicular body formation), autophagy acts as a degradation system for long-lived proteins and damaged organelles, often relying on ESCRT components for phagophore sealing and autophagosome-lysosome fusion. Autophagy begins with the nucleation and formation of a double-membrane structure, the phagophore, which gradually expands to engulf cellular proteins and organelles [18]. This process can be induced by nutrient deprivation, DNA damage, hypoxia, endoplasmic reticulum (ER) stress, and infection [4]. Stress-induced autophagy is primarily regulated by AMP-activated protein kinase (AMPK) activation and mammalian target of rapamycin (mTOR) inhibition [19]. The initiation of autophagy in mammalian cells relies on the unc-51-like kinase (ULK) complex, which includes the ULK1 (or ULK2) protein kinase, FIP200, autophagy-related (ATG) gene 13 (ATG13), and ATG10. Upon activation, the ULK1 complex stimulates the class III phosphatidylinositol 3-kinase complex I (PI3KC3-C1), consisting of VPS34, VPS15, BECN1, and ATG14. Phosphatidylinositol-3-phosphate (PI(3)P) is then generated [5]. It facilitates the formation of omegasomes-specialized structures on the ER membrane, which act as nucleation sites for autophagosome development [20]. Additional membrane sources for phagophore formation include the outer mitochondrial membrane [21] and the plasma membrane [22]. As the phagophore grows and engulfs substrates, various ATG proteins are recruited, including the lipidation of light chain 3 (LC3), which marks the maturation into a closed autophagosome. In the final stage, the autophagosome fuses with a lysosome to form an autolysosome, where the enclosed cargo is degraded. This dynamic process is essential for cellular homeostasis, recycling cellular components, and responding to stress conditions.

## 4. Crosstalk Between Autophagy and ESCRT Pathways

The role of ESCRT machinery in both MVB and autophagosome formation, as well as its involvement in the fusion of MVBs with autophagosomes, has become clearer with the advent of new technologies. These advancements have provided valuable insights into how ESCRT components contribute to the coordination between these two degradation pathways. Table 2 and Figure 1 summarize ESCRT functions in autophagy induction signals and membrane dynamics. For instance, HRS depletion induces ER stress and autophagy [23]. ESCRT regulates coat protein complex II (COPII) vesicle transport and ER-Golgi intermediate compartment (ERGIC) assembly, facilitating autophagosome formation [24]. ESCRT is also indispensable in autophagosome closure and sealing. Electron microscopy often struggles to distinguish between fully closed and unclosed autophagosomes, but innovative techniques have provided deeper insights into this process. For instance, the HaloTag-LC3 autophagosome completion assay has demonstrated that inhibiting CHMP2A and VPS4 prevents proper phagophore closure and sealing, leading to an accumulation of unclosed autophagic membranes [25]. Similarly, a novel optogenetic closure assay has confirmed that CHMP2A and CHMP4B depletion inhibit phagophore sealing [26]. In yeast, the Rab5 GTPase Vps21 coordinates the assembly of Snf7 and the Vps4 at the phagophore assembly site, facilitating phagophore expansion and subsequent sealing of the autophagosome, ensuring cargo encapsulation [27]. These findings collectively underscore the critical role of ESCRT components in autophagosome completion. This incomplete closure prevents autophagosome fusion with lysosomes, resulting in the accumulation of autophagosomes and the retention of undegraded substrates [28]. Depletion or gene mutants of VPS4/SKD1, a mammalian homologue of yeast Vps4 [29], CHMP2B [30], the mouse CHMP4 homolog mSnf7 [31], and VPS4 [25,32] consistently indicate significant impairment in autophagosome transport to lysosomes and degradation. These findings collectively underscore the critical role of ESCRT components in autophagosome closure and sealing. Alternatively, autophagosomes can fuse with MVBs to create an intermediate compartment called an amphisome, which is one of the mechanisms that allow autophagosomes to further mature before fusion with lysosomes [33]. Therefore, ESCRT dysfunction may also impair amphisome formation. Interestingly, the protein Alix presents a nuanced role, with its absence impairing basal autophagic flux through interactions with the ATG12-ATG3 complex [34].

## 5. ESCRT and Lysosome

Lysosome biogenesis is essential for cellular degradation and recycling, yet the precise role of ESCRT in lysosome activity remains only partially understood. TSG101 and VPS28 play key roles in regulating lysosome size, facilitating the degradation of proteins from lysosomal membranes, and maintaining cellular homeostasis [35]. ESCRT also functions in lysosomal membrane repair. Lysosomotropic agents like L-leucyl-L-leucine methyl ester or silica crystals lead to damaged lysosomes. ESCRT-I and Alix are the first to respond to lysosomal damage, followed by the recruitment of ESCRT-III and VPS4, which collaborate to seal membrane lesions [36]. This repair mechanism may share similarities with intraluminal vesicle biogenesis, although the molecular details remain unclear. These observations underscore ESCRT’s essential role in lysosome activity.

## 6. ESCRT in Diseases

As ESCRT machinery is essential for maintaining normal cellular functions, its dysfunction is associated with various types of diseases.

ESCRT dysfunction is found in an increasing number of neurodegenerative disorders. Impaired ESCRT components, such as HRS [23], CHMP2B [37], and VPS4A [38], significantly impair autophagic, endosomal, and lysosomal degradation pathways, exacerbating the aggregation of amyloid-beta (Aβ) and neuronal damage. Compounds such as NCT-504 promote the degradation of aggregation-prone proteins through CHMP4A/VPS4A-mediated ESCRT-MVB pathways, suggesting that targeted modulation of this degradation axis may provide novel therapeutic strategies against neurodegenerative progression [39].

Dysregulation of the ESCRT pathway has been increasingly implicated in cancer progression. The components’ roles as oncogenic drivers or tumor suppressors depend heavily on cellular context. Specifically, HRS (ESCRT-0) and TSG101 (ESCRT-I) are upregulated in cancers and promote tumor progression [19]. Inhibition of HRS or TSG101 limits tumor growth [40,41,42]. CHMP1A (ESCRT-III) inhibits tumor growth in pancreatic cancer [43], while CHMP4C (ESCRT-III) safeguards genomic stability [44]. VPS4 plays essential roles in cell division, metastasis, cell death, and signaling cascades, emphasizing its multifaceted role in cancer biology [45]. Collectively, these findings highlight ESCRT components as promising diagnostic markers and therapeutic targets in oncology.

## 7. ESCRT and HBV

The budding of enveloped viruses, such as human immunodeficiency virus (HIV-1), Ebola virus, and HBV, closely resembles intraluminal vesicle formation, relying on ubiquitin, ESCRT recruitment, and VPS4 activity [8,46,47,48,49]. Overall, the viral budding process involves four key steps: (1) transport of viral proteins to MVBs; (2) recruitment of ESCRT-I and Alix; (3) engagement of ESCRT-II and ESCRT-III at the viral particle-host membrane interface; (4) VPS4-mediated disassembly and recycling of ESCRT-III subunits [50]. This mechanism facilitates efficient viral replication and release from infected host cells.

Our group has established the essential role of the autophagy-MVB axis in orchestrating HBV replication, assembly, and intracellular trafficking. Building on these discoveries, this review synthesizes our mechanistic findings with emerging evidence on HBV’s exploitation of ESCRT machinery, revealing novel viral hijacking mechanisms through membrane remodeling processes.

HBV is an hepatotropic enveloped DNA virus that replicates via a reverse transcription process. The viral genome contains four overlapping open reading frames, encoding the polymerase/reverse transcriptase (RT), the capsid-forming core protein and its related secretory precore protein, three related envelope proteins (small, SHBs; middle, MHBs; large, LHBs), and the regulatory X protein [51]. For progeny virus formation, newly synthesized core monomers assemble into capsids in which reverse transcription takes place. Upon maturation, nucleocapsids are next enveloped through their interaction with LHBs and are released as mature virions. Additionally, manifold particle types, including non-infectious empty virions, empty spherical and filamentous envelope particles, and naked capsids, are released that contribute to the extremely successful spread of HBV [52]. The assembly and release of HBV subviral particles occur through the ER-Golgi pathway, whereas virion assembly and release involve the endosomal pathway and autophagy [8,49].

The interplay between HBV and the ESCRT machinery has been studied for almost two decades. The regulation of the HBV life cycle by the ESCRT machinery is highly complex. As illustrated in Figure 2, a concise summary is provided of the effects observed in the Huh7 cell line following siRNA-mediated knockdown of ESCRT components or overexpression of dominant-negative ESCRT mutants on the secretion of HBV virions, capsids, and subviral particles (SVPs). This experimental approach highlights the critical role of ESCRT in distinct stages of HBV assembly and egress, particularly in modulating the trafficking and release of viral and subviral components. Appendix A synthesizes two decades of research delineating how modulating ESCRT expression (via siRNA, plasmid overexpression, or dominant mutants) impacts intracellular and extracellular HBV RNA, HBV DNA, capsid, and HBsAg levels. This comprehensive dataset highlights the spatiotemporal regulation of HBV component trafficking by ESCRT machinery, pinpointing its dual role in both viral particle assembly and selective release of subviral components.

The first evidence for the importance of ESCRTs in HBV virion secretion was obtained with mutants of the AAA ATPase VPS4A/B (K173Q and E228Q) that are defective in ATP binding and hydrolysis, respectively. The overexpression of the mutants nearly completely inhibits HBV replication and HBV virion secretion in hepatoma cell lines [53]. HBV budding also requires scission functions provided by ESCRT-III (CHMP3, CHMP4B/C), as shown by overexpression and depletion studies [54,55]. Similarly, interferences with VPS4 and ESCRT function significantly reduce the release of filamentous subviral envelope particles (fSPVs) but have minimal impacts on the secretion of spherical SVPs (sSVPs) [54,55,56]. These differential effects suggest that the viral particle types are released through distinct pathways.

Studies on the involvement of ESCRT-II in HBV replication are limited. The depletion of ESCRT-II components (EAP20, EAP30, and EAP45) significantly reduces HBV virion egress without affecting intracellular core or HBsAg levels. Notably, the depletion of EAP30 and EAP45 reduced intracellular capsids and pregenomic RNA (pgRNA), suggesting a potential role in HBV nucleocapsid formation and encapsidation [57]. However, these findings require further validation.

The ESCRT-I component TSG101 also plays a complex role in HBV egress, although the findings are contradictory thus far. One study reported that the depletion of TSG101 increases HBV virion secretion by destabilizing the ESCRT-I complex and reducing VPS28 levels [57]. However, another study could not detect any role for TSG101 in HBV production [58]. In complete contrast, TSG101 was reported to recognize the PPAY motif of ubiquitinated HBV core/capsid protein and facilitate viral recruitment to MVBs. In this context, the depletion of TSG101 reduces HBV secretion and increases intracellular capsid accumulation in HepG2-NTCP and HepAD38 cells [59]. Further studies are needed to clarify the exact way in which HBV enters the ESCRT pathway.

Beyond that, siRNA screening identified the ESCRT-0 complex as a critical checkpoint regulating different egress pathways of HBV particle types. Aberrant levels of the ESCRT-0 subunit HGS/HRS, caused by its depletion and overexpression, suppress HBV transcription, replication, and virion secretion. However, HGS/HRS depletion reduces naked capsid secretion, whereas overexpression enhances it [58].

An inverse correlation between secreted virions and secreted naked capsids was likewise observed upon functional inactivation of ESCRT-III and VPS4 [56], demonstrating that HBV naked capsid budding does not require either ESCRT-0 or a functional ESCRT scission machinery. More evidence shows that Alix and its Bro1 domain are required for HBV naked capsid secretion. Ectopic overexpression of Alix and even its Bro1 domain enhances capsid egress, while its depletion consistently has opposite effects [56,60]. A screening of NEDD4 E3 ubiquitin ligase family members identified AIP4 as an interaction partner of Alix that promotes Alix-mediated naked capsid secretion.

Besides ESCRT and ESCRT-associated proteins, MVBs are likely to participate in HBV virion release. CD63 is a prominent marker and essential component of MVBs. The depletion of CD63 leads to the accumulation of intracellular LHBs and a significant reduction in the release of infectious HBV particles [61].

ESCRT proteins play a critical role in the formation of MVBs and autophagosomes, both of which are involved in the HBV lifecycle. This raises the question of whether ESCRT proteins’ role extends beyond direct viral interactions, encompassing endosomal and autophagic pathways. Our preliminary unpublished data demonstrate this complex interplay: silencing Alix impairs autophagosome formation, subsequently compromising HBV assembly and reducing capsid secretion. Similarly, expressing a dominant-negative VPS4A mutant induces abnormal lysosomal activities, leading to increased viral degradation. These findings suggest that abnormal ESCRT expression can significantly disrupt endosomal and autophagic processes, as well as lysosomal activities, thereby interfering with key stages of the HBV lifecycle, particularly viral assembly and secretion. Understanding HBV’s interaction with ESCRT requires a comprehensive approach that considers both the proteins’ direct viral binding functions and their broader impacts on cellular biological activities. Future studies should simultaneously examine these multifaceted interactions to fully elucidate the mechanisms governing viral lifecycle and cellular response.

## 8. Conclusions

Traditionally, ESCRTs have been recognized for their role in the formation and maturation of MVBs. However, recent findings have uncovered a broader connection between ESCRT components and autophagy, linking ESCRT dysfunction to a range of human diseases. Specifically, ESCRT-dependent processes involved in the closure and maturation of endosomes and autophagosomes are critical for their fusion with lysosomes. This fusion is essential for proper cargo sorting and degradation regulation. In this review, we have highlighted the key roles of the ESCRT machinery in MVB formation, autophagy, and the HBV lifecycle, emphasizing its integrated functions in both autophagic and endosomal pathways. Several key questions remain to be addressed. First, how do autophagosomes recognize specific cargo and sequentially recruit ESCRT components? Second, although viruses such as HBV exploit ESCRT machinery and depend on autophagy for replication and secretion, the precise mechanisms by which ESCRTs facilitate viral entry into autophagosomes are not fully understood. Finally, VPS4, which mediates the disassembly of the ESCRT complex, may have a specialized role in detaching ESCRT from autophagosomes. Addressing these questions will provide valuable insights into the regulatory roles of ESCRT and autophagy in both health and disease.

## Figures and Tables

**Figure 1 cells-14-00603-f001:**
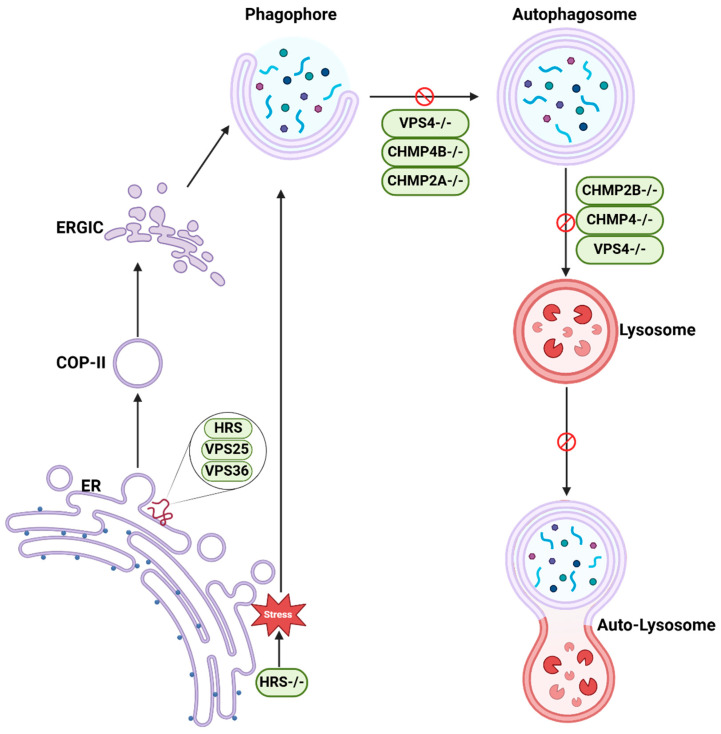
The roles of ESCRT in autophagy. The ESCRT machinery is critical for regulating various stages of autophagy. Disruption of HRS (HRS^−/−^) triggers ER stress, which activates autophagy. HRS, VPS25, and VPS36 contribute to COP-II vesicle transport, ER-Golgi intermediate compartment (ERGIC) assembly, and autophagosome formation. During autophagosome maturation, the inhibition of ESCRT-III components such as CHMP2 and CHMP4, as well as VPS4, disrupts the closure and sealing of the phagophore. This impairment prevents the fusion of autophagosomes with lysosomes, resulting in the accumulation of unclosed autophagic membranes. These findings underscore the essential role of ESCRT in maintaining autophagic flux and cellular homeostasis. Endoplasmic Reticulum: ER; Coat Protein Complex II: COP-II; Endoplasmic Reticulum-Golgi Intermediate Compartment: ERGIC. The figure was created in Biorender (https://biorender.com/).

**Figure 2 cells-14-00603-f002:**
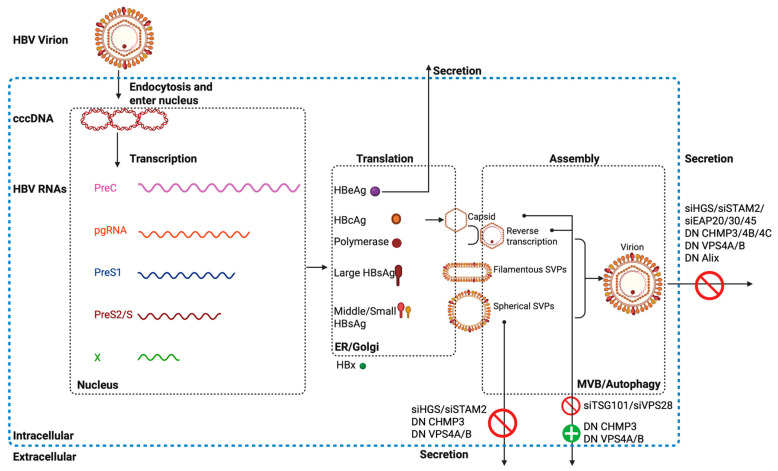
Schematic of ESCRT-mediated modulation of the HBV particle secretion. HBV virions bind and enter host cells, followed by the conversion of relaxed circular DNA (rcDNA) into covalently closed circular DNA (cccDNA) within the nucleus. The cccDNA serves as a transcriptional template to synthesize viral RNAs (depicted as wavy lines), including: PreC mRNA, pregenomic RNA (pgRNA), PreS1 mRNA, PreS2/S mRNA, and X mRNA. These transcripts encode distinct viral proteins: PreC mRNA for HBeAg; pgRNA for the HBcAg and polymerase; PreS1 mRNA for the Large HBsAg; PreS2/S mRNA for the Medium and Small HBsAg. Reverse transcription occurs within newly assembled capsids, generating rcDNA-containing nucleocapsids. Mature HBV virions (enveloped particles containing rcDNA), subviral particles (SVPs; spherical and filamentous forms exclusively containing HBsAg), and capsids (with or without rcDNA) are subsequently secreted. Notably, experimental inhibition of specific ESCRT components (via siRNA knockdown or dominant-negative mutants) disrupts the secretion of these viral and subviral structures are summarized. The figure was created in Biorender.

**Table 1 cells-14-00603-t001:** ESCRT component nomenclature across mammals, yeast (*S. cerevisiae*), and *Drosophila*.

Complex	Mammals	Yeast (*S. cerevisiae*)	*Drosophila*
ESCRT-0	HGS (HRS), STAM1/2	Vps27, Hse1	Hrs, Stam
ESCRT-I	TSG101, VPS28, VPS37A/B/C/D/, MVB12	Vps23, Vps28, Vps37, MVB12	Tsg101, Vps28, Vps37
ESCRT-II	EAP45, EAP30, EAP20	Vps22, Vps25, Vps36	Vps22, Vps25, Vps36
ESCRT-III	CHMP1A/B, CHMP2A/B, CHMP3, CHMP4A/B/C, CHMP5, CHMP6, CHMP7, CHMP8/IST1	Did2, Vps2, Vps24, Snf7, Vps60, Vps20, Chm7, Ist1	Shrub (CHMP4), CHMP2, CHMP3, CHMP1, IST1
VPS4 ATPase System	VPS4A/B, LIP5	Vps4, Vta1	Vps4, CG7913 (Vta1-like)
Accessory Proteins	Alix (PDCD6IP)	Bro1, Doa4	Alix (homolog?), Bro1

Hepatocyte growth factor-regulated tyrosine kinase substrate: HGS/HRS; Signal Transducing Adaptor Molecule: STAM1/2; Tumor Susceptibility Gene 101: TSG101; Vacuolar Protein Sorting: VPS; Multivesicular Body: MVB; Charged Multivesicular Body Protein: CHMP; ELL-Associated Protein: EAP; Lysosomal trafficking regulator Interacting protein 5: LIP5; Alix: Apoptosis-linked gene 2 (ALG-2)-interacting protein X; Vacuolar transport associated 1: Vta; Hse1: Hrs-binding protein with ubiquitin isopeptidase activity 1; Did2: Degradation in the vacuole defective 2; IST1: Increased sodium tolerance 1; Doa4: Degradation of alpha 4.

**Table 2 cells-14-00603-t002:** The Roles of ESCRT in Autophagy.

Target	Model	Results	Conclusion	Reference
HRS	Primary hippocampal neurons and mice brain	HRS-deficient cells induce ER stress activation and subsequent JNK signaling and increase LC3 and p62.	Silencing of HRS impairs the late stage of autophagic flux.	[23]
Hrs Vps25/32/13D	Drosophila intestine cells	Cells lacking either ESCRTs influence ER maturation, COPII trafficking, ER-Golgi intermediate compartment assembly, and autophagosome formation.	ESCRTs regulates COPII vesicle formation that influences autophagy.	[24]
CHMP2A	HeLa cells and U-2 OS cells	CHMP2A deficiency results in phagophore accumulation.	CHMP2A translocates to the phagophore and regulates the separation of the inner and outer autophagosomal membranes to form double-membrane autophagosomes.	[25]
CHMP2A/4B	Human retinal pigment epithelial cells	CHMP4B is recruited transiently to nascent autophagosomes, and depletion of CHMP2A inhibited phagophore sealing during mitophagy.	CHMP2A and CHMP4B mediate phagophore closure.	[26]
Snf7 Vps4	Yeast cells	Depletion of Snf7 and the Vps4 causes late autophagy defects and accumulation of autophagosomes.	Rab5 controlls Atg17-Snf7 interaction leading to recruitment of ESCRT to open autophagosomes and catalyzing their closure.	[27]
VPS4	Mouse	VPS4/SKD1 dominant-negative mutant causes a defect in autophagy-dependent bulk protein degradation.	VPS4/SKD1 is required for formation of autolysosomes.	[29]
CHMP2B	Neuron cells	CHMP2B mutants lead to accumulation of protein aggregates.	Efficient autophagic degradation requires functional MVBs.	[30]
Snf7 CHMP2B	Cortical neurons and flies	The loss of mSnf7-2 or CHMP2B(Intron5) expression cause the accumulation of autophagosomes.	ESCRT-III dysfunction is associated with the autophagy pathway.	[31]
VPS27/32 VPS4	*Caenorhabditis elegans*	All ESCRT mutants present an accumulation of abnormal endosomes and autophagosomes.	The accumulation of autophagosomes is secondary to the formation of enlarged endosomes and is due to the induction of the autophagic flux.	[32]
VPS4	HeLa cells	The inhibition of the AAA-ATPase VPS4 activity impairs autophagosome completion.	ESCRT machinery acts in the final step of autophagosome formation.	[25]
Alix	Mouse embryonic fibroblasts	Alix depletion leads to a reduction in basal autophagy.	The interactions between ATG12-ATG3 and Alix promote basal autophagy.	[34]

## Data Availability

Not applicable.

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
