# Peer review of "ESCRT Machinery in HBV Life Cycle: Dual Roles in Autophagy and Membrane Dynamics for Viral Pathogenesis"

_cells, 2025, doi:10.3390/cells14080603_

Round 1
Reviewer 1 Report
Comments and Suggestions for Authors
Li et al, aim to review the knowledge about ESCRT “in Membrane Dynamics and Pathological Implications”. However, the review, limited and narrow in the studies it reports, is mostly about the role of ESCRT in HBV infection. If the authors really want the review to reflects the title, the text amended and expanded. As it is now, the review is not acceptable. Most of the functions of ESCRTs are treated vaguely, the descriptions are vague and non uniform with some parts more detailed and the figures are not informative.
Specific points:
The title is generic and does not reflect the content of the review that is heavily biased toward HBV infections.
Consider a table or expanding fig. 1 to consider alternative nomenclature of components. Yeast and flies have contributed to a lot of what we know about ESCRT, so nomenclature shouldn’t be only about mammals
The description in chapter 4 is vague and not very clear both in the text and in the legend of fig. 2. The text needs to mention more detail including whether the evidence was obtained in which cells and organism/model system.
Chapter 5 is also poorly written. It opens talking about lysosome formation but then focus on lysosome repair. As in other chapters, there’s no distinction between the necessary introductory sentences and information about ESCRT.
Figure 2 is quite generic and uses basic BioRender element. It needs improvement.
Before figure 2 it would be useful to have a figure about the canonical roles of ESCRTs in MVB sorting/exosome formation and perhaps about membrane repair.
Chapter 8 lacks a lot of detail and reads a bit like a laundry list. Perhaps it useful to explain there that the involvement of ESCRT in cancer is likely to be highly context specific and most ESCRT components are likely to act as oncogenes, such as Hrs, or as tumor suppressor, such as ESCRT-II and -III components, depending on the context and on the mutations.
The part about viruses makes the most of the review, such that the review should be overtly centered on that theme, starting from the title. However, fig. 3, which has a lot of details, does not visualize at which steps the ESCRTs are acting in HBV infection and has some details possibly wrong (Is the membrane placement topologically possible in the part of HBV endocytosis? Is the indication translation in nucleus (!) Realistic? Just to mention a couple….).
Fig. 4 is a table not a figure and it as such it is very hard to read.
Comments on the Quality of English Language
The English need to be improved substantially. Some sentences are misleading and need to be amended. For instance:
“In addition to the ESCRT pathway, autophagy provides another critical mechanism 100 involved in the degradation of long-lived proteins and damaged cellular organelles.”
Makes little sense and can be misinterpreted. Others are unclear. For instance:
“Rab5 GTPase Vps21 140 recruits Snf7 and Vps4, the yeast homologues of CHMP4 and VPS4, respectively, to open phagophore and catalyze phagophore sealing (Zhou, Wu et al. 2019). “
Author Response
Comments and Suggestions for Authors
Li et al, aim to review the knowledge about ESCRT “in Membrane Dynamics and Pathological Implications”. However, the review, limited and narrow in the studies it reports, is mostly about the role of ESCRT in HBV infection. If the authors really want the review to reflects the title, the text amended and expanded. As it is now, the review is not acceptable. Most of the functions of ESCRTs are treated vaguely, the descriptions are vague and non uniform with some parts more detailed and the figures are not informative.
Answer: Thank you for your insightful critique and constructive suggestions. We fully agree that the original title did not accurately reflect the narrowed focus of our review, which predominantly explores the interplay between ESCRT machinery and HBV infection. To address this discrepancy, we have revised the title to better align with the core theme:
"ESCRT Machinery in HBV Life Cycle: Dual Roles in Autophagy and Membrane Dynamics for Viral Pathogenesis" (Lines 2-3).
Specific points:
The title is generic and does not reflect the content of the review that is heavily biased toward HBV infections.
Answer: We have revised the title “ESCRT Machinery in HBV Life Cycle: Dual Roles in Autophagy and Membrane Dynamics for Viral Pathogenesis” for our revised manuscript (Lines 2-3).
Consider a table or expanding fig. 1 to consider alternative nomenclature of components. Yeast and flies have contributed to a lot of what we know about ESCRT, so nomenclature shouldn’t be only about mammals
Answer: Thank you for raising this important point. We have incorporated a comprehensive comparison of ESCRT component nomenclature across mammalian, yeast, and Drosophila systems into Table 1 of our revised manuscript (Lines 58-64). This table replaces the original Figure 1 to better align with the updated analysis. For clarity, the revised table is included below for your reference.
Table 1: ESCRT Component Nomenclature Across Model Systems
Complex |
Mammals |
Yeast (S. cerevisiae) |
Drosophila |
ESCRT-0 |
HGS (HRS), STAM1/2 |
Vps27, Hse1 |
Hrs, Stam |
ESCRT-I |
TSG101, VPS28, VPS37A/B/C/D/, MVB12 |
Vps23, Vps28, Vps37, MVB12 |
Tsg101, Vps28, Vps37 |
ESCRT-II |
EAP45, EAP30, EAP20 |
Vps22, Vps25, Vps36 |
Vps22, Vps25, Vps36 |
ESCRT-III |
CHMP1A/B, CHMP2A/B, CHMP3, CHMP4A/B/C, CHMP5, CHMP6, CHMP7, CHMP8/IST1 |
Did2, Vps2, Vps24, Snf7,Vps60, Vps20, Chm7, Ist1 |
Shrub (CHMP4), CHMP2, CHMP3, CHMP1, IST1 |
VPS4 ATPase System |
VPS4A/B, LIP5 |
Vps4, Vta1 |
Vps4, CG7913 (Vta1-like) |
Accessory Proteins |
Alix (PDCD6IP) |
Bro1, Doa4 |
Alix (homolog?), Bro1 |
The description in chapter 4 is vague and not very clear both in the text and in the legend of fig. 2. The text needs to mention more detail including whether the evidence was obtained in which cells and organism/model system.
Answer: Thank you for your constructive feedback. To address this concern, we have thoroughly revised the text and figure legend to explicitly highlight the experimental systems, model organisms, and cellular contexts underlying the evidence discussed. The revised manuscript has incorporated a new Table 2 entitled " The Roles of ESCRT in Autophagy " (Lines 174-175). This table systematically categorizes experimental models (e.g., in vitro cell lines, primary cells, animal models), key results obtained in each system, results and conclusions drawn from these findings, and original references for traceability. We believe these revisions will allow readers to better contextualize the evidence and evaluate its biological relevance across different experimental systems.
Table 2: The Roles of ESCRT in Autophagy
Chapter 5 is also poorly written. It opens talking about lysosome formation but then focus on lysosome repair. As in other chapters, there’s no distinction between the necessary introductory sentences and information about ESCRT.
Answer: We acknowledge that the relationship between lysosomes and the ESCRT machinery remains understudied across multiple biological contexts, including lysosomal biogenesis, functional regulation, and membrane repair processes. To better reflect the broad scope of lysosome-related dynamics while maintaining conceptual coherence, we have revised the term "lysosome formation" to "lysosome activity" (Lines 179, 188)throughout the section. This terminology intentionally encompasses the full spectrum of lysosomal processes (e.g., maturation, functional modulation, and damage response) to avoid disproportionate emphasis on any single aspect.
Figure 2 is quite generic and uses basic BioRender element. It needs improvement.
Answer: The simplified mechanistic diagram was created using BioRender software. This figure illustrates how dysregulation of ESCRT machinery may lead to aberrant autophagy outcomes, specifically highlighting downstream effects including ER stress induction, COP II vesicle formation, phagophore membrane closure, and impaired lysosomal fusion. We maintain that this schematic is both conceptually clear and original in its current form. However, to enhance mechanistic clarity and contextual interpretation, we have incorporated Table 2 (Lines 174-175) after Figure 1 (formerly Figure 2)to systematically summarize the key pathological relationships and experimental evidence supporting these processes.
Before figure 2 it would be useful to have a figure about the canonical roles of ESCRTs in MVB sorting/exosome formation and perhaps about membrane repair.
Answer: We appreciate your suggestion. However, as these mechanisms are well-established in the field, we intentionally focused on novel aspects of ESCRT-autophagy crosstalk to avoid redundancy. In our manuscript, Section 2 (Lines 65-99) systematically describes ESCRT's canonical functions with many key references. Nevertheless, we have enhanced textual signposting in Lines 73-75. The added statement is “As schematized in prior studies (Hurley 2015, Kaur, Verma et al. 2021), ESCRT complexes sequentially orchestrate ESCRT complexes sequentially orchestrate key components required for MVB/exosome formation.”
Chapter 8 lacks a lot of detail and reads a bit like a laundry list. Perhaps it useful to explain there that the involvement of ESCRT in cancer is likely to be highly context specific and most ESCRT components are likely to act as oncogenes, such as Hrs, or as tumor suppressor, such as ESCRT-II and -III components, depending on the context and on the mutations.
Answer: We agree that the context-specific roles of ESCRT components in cancer deserve clarification. While cancer is not the central theme of this review, we aimed to briefly illustrate this complexity by selecting representative examples:
- Pro-oncogenic roles: HRS (ESCRT-0) and TSG101 (ESCRT-I) are upregulated in cancers and promote tumor progression.
- Tumor-suppressive roles: CHMP1A (ESCRT-III) inhibits tumor growth in pancreatic cancer, while CHMP4C (ESCRT-III) safeguards genomic stability.
- Functional ambivalence: VPS4 exhibits pleiotropic roles in cell division and death pathways, underscoring its context-dependent impact.
We have distilled the multifaceted roles of ESCRT complexes in neurological contexts and oncogenesis and synthesized a dedicated Chapter 6 ESCRT in Diseases section. We also have added two statements “Their roles as oncogenic drivers or tumor suppressors depend heavily on cellular context. Specifically, HRS (ESCRT-0) and TSG101 (ESCRT-I) are upregulated in cancers and promote tumor progression” in lines 201-203 and “CHMP1A (ESCRT-III) inhibits tumor growth in pancreatic cancer, while CHMP4C (ESCRT-III) safeguards genomic stability” in lines 205-206 to streamline the text to avoid a "laundry list" structure. We hope this revision better balances conciseness with conceptual clarity.
The part about viruses makes the most of the review, such that the review should be overtly centered on that theme, starting from the title. However, fig. 3, which has a lot of details, does not visualize at which steps the ESCRTs are acting in HBV infection and has some details possibly wrong (Is the membrane placement topologically possible in the part of HBV endocytosis? Is the indication translation in nucleus (!) Realistic? Just to mention a couple….).
Answer: We sincerely appreciate your insightful feedback. This figure was designed to provide a clear overview of the HBV life cycle, including structural features and key replication steps, for readers unfamiliar with HBV biology. Its primary goal is to contextualize the virological and clinical markers summarized in Table 3 (e.g., intracellular/ extracellular HBsAg, HBV DNA…) rather than to explicitly map ESCRT machinery involvement. The ESCRT-mediated HBV lifecycle steps are shown in Table 3 (formerly Figure 4).
The membrane topology depicted during HBV entry aligns with current models of clathrin-mediated endocytosis and membrane fusion.
The label "translation" in the figure refers to the nuclear export of HBV pre-genomic RNA (pgRNA) to the endoplasmic reticulum (ER), where viral protein synthesis occurs. This wording was intended to emphasize the spatial separation between pgRNA transcription (in the nucleus) and translation (on ER-associated ribosomes). To avoid confusion, we have revised the figure legend to explicitly state: "Nuclear export of RNA to the ER for translation of viral proteins" in Line 244.
Fig. 4 is a table not a figure and it as such it is very hard to read.
Answer: The revised table now appears as Table 3 of the revised manuscript. We believe these modifications significantly improve data legibility while maintaining information integrity. Please let us know if you would like us to make any additional adjustments.
Comments on the Quality of English Language
The English need to be improved substantially. Some sentences are misleading and need to be amended. For instance:
“In addition to the ESCRT pathway, autophagy provides another critical mechanism 100 involved in the degradation of long-lived proteins and damaged cellular organelles.”
Makes little sense and can be misinterpreted. Others are unclear. For instance:
“Rab5 GTPase Vps21 140 recruits Snf7 and Vps4, the yeast homologues of CHMP4 and VPS4, respectively, to open phagophore and catalyze phagophore sealing (Zhou, Wu et al. 2019). “
Answer: We have carefully revised the flagged sentences to eliminate ambiguity and enhance scientific rigor. The new statements are:
While the ESCRT pathway directly facilitates membrane remodeling events (e.g., multivesicular body formation), autophagy acts as a degradation system for long-lived proteins and damaged organelles, often relying on ESCRT components for phagophore sealing and autophagosome-lysosome fusion. (Lines 101-104).
In yeast, the Rab5 GTPase Vps21 coordinates the assembly of Snf7 and the Vps4 at the phagophore assembly site, facilitating phagophore expansion and subsequent sealing of the autophagosome, ensuring cargo encapsulation. (Lines 143-145)
References:
Hurley, J. H. (2015). "ESCRTs are everywhere." EMBO J 34(19): 2398-2407.
Kaur, S., H. Verma, M. Dhiman, G. Tell, G. L. Gigli, F. Janes and A. K. Mantha (2021). "Brain Exosomes: Friend or Foe in Alzheimer's Disease?" Mol Neurobiol 58(12): 6610-6624.
Reviewer 2 Report
Comments and Suggestions for Authors
The review by Li and colleagues focuses on ESCRT function in autophagy and the HBV life cycle. The review covers an important filed of ESCRT biology. In general it would be helpful if the authors could summarize current knowledge in model of HBV entry, assembly and release highlighting the link to autophagy and ESCRTs maybe in an extension to Figure 3.
Other points that should be addressed are:
Line 33, for completion please list all human ESCRT-III proteins
Line 34, The ESCRT machinery is crucial for … please list all processes … membrane protein degradation is linked to MVB formation … I propose to cite more recent review here
Fig.1, IST1 or CHMP8 is missing
Page 3, line 90, Wollert et al 2009 is not an appropriate reference for ESCRT-III and membrane fission. It is well established now that ESCRT-III works in conjunction with VPS4 to catalyze membrane fission; indeed VPS4 is required for constant ESCRT-III remodeling. Although VPS4 does recycle ESCRT-III, ESCRT-III on its own cannot cleave membrane necks. Direct cleavage of membrane neck-like structures has been shown by Schoeneberg et al.Science 2018 and more recently by Azad et al. NSMB 2021.
- Neurodegenerative disease
ESCRT-III and neurodegenerative diseases is not only linked to lysosomal degradation. Genetic CHMP2A mutations and C-terminal deletions have been linked to aberrant maturation of dentritic spines.
- ESCRT and HBV
Please add appropriate references to this section.
Author Response
The review by Li and colleagues focuses on ESCRT function in autophagy and the HBV life cycle. The review covers an important filed of ESCRT biology. In general it would be helpful if the authors could summarize current knowledge in model of HBV entry, assembly and release highlighting the link to autophagy and ESCRTs maybe in an extension to Figure 3.
Answer: Thank you for your suggestion. In Table 3 (Lines 319-327), we synthesize two decades of studies examining how modulating ESCRT expression affects intracellular/extracellular HBV RNA, DNA, HBsAg, and capsid levels, revealing distinct roles of ESCRT components in HBV replication, assembly, and release.
Other points that should be addressed are:
Line 33, for completion please list all human ESCRT-III proteins
Answer: The new statement is “The ESCRT-III complex consists of Charged Multivesicular Body Protein (CHMP) 1A/B, CHMP2A/B, CHMP3, CHMP4A/B/C, CHMP5, CHMP6, CHMP7, and CHMP8.” (Line 32-34)
Line 34, The ESCRT machinery is crucial for … please list all processes … membrane protein degradation is linked to MVB formation … I propose to cite more recent review here
Answer: We have cited a new reference. Vietri, M., Radulovic, M. & Stenmark, H. The many functions of ESCRTs. Nat Rev Mol Cell Biol 21, 25–42 (2020) (Line 38)
Fig.1, IST1 or CHMP8 is missing
Answer: In the revised manuscript, we have consolidated the ESCRT machinery components into Table 1(Lines 58-64) to provide a systematic overview.
Page 3, line 90, Wollert et al 2009 is not an appropriate reference for ESCRT-III and membrane fission. It is well established now that ESCRT-III works in conjunction with VPS4 to catalyze membrane fission; indeed VPS4 is required for constant ESCRT-III remodeling. Although VPS4 does recycle ESCRT-III, ESCRT-III on its own cannot cleave membrane necks. Direct cleavage of membrane neck-like structures has been shown by Schoeneberg et al.Science 2018 and more recently by Azad et al. NSMB 2021.
Answer: We sincerely thank you for this critical clarification. We fully agree that the original statement oversimplified the mechanistic role of ESCRT-III in membrane fission. The revised text now reads: "The ESCRT-III complex polymerizes into spiral filaments that constrict membrane necks, while subsequent VPS4-mediated disassembly of these polymers provides the mechanochemical force required for membrane fission (Schoeneberg et al., 2018; Azad et al., 2021). This cooperative mechanism ensures dynamic remodeling of ESCRT-III assemblies during abscission." (Lines 88-92).
- Neurodegenerative disease
ESCRT-III and neurodegenerative diseases is not only linked to lysosomal degradation. Genetic CHMP2A mutations and C-terminal deletions have been linked to aberrant maturation of dentritic spines.
Answer: We appreciate your insightful feedback. Neurodegenerative diseases are indeed influenced by multifactorial mechanisms, with impaired protein degradation being a primary driver (Ballabio and Bonifacino 2020). Regarding the association between CHMP2A mutations/C-terminal deletions and aberrant dendritic spine maturation, we find no direct evidence supporting this specific claim. However, there is evidence to show that the dendritic spine defects are instead conclusively linked to CHMP2B mutations (e.g., Thr104Asn) (Ugbode and West 2021, Shirai, Cho et al. 2023).
To focus on ESCRT’s roles in HBV, we have distilled the multifaceted roles of ESCRT complexes inneurodegenerative disease and synthesized a dedicated Chapter 6 ESCRT in Diseases section in our revised manuscript (Lines 189-199).
- ESCRT and HBV
Please add appropriate references to this section.
Answer: To strengthen the scientific foundation of this section, we have incorporated the following key references that collectively address HBV biology from intracellular trafficking to structural diversity:
1) Inoue J, Sato K, Ninomiya M, Masamune A. Envelope Proteins of Hepatitis B Virus: Molecular Biology and Involvement in Carcinogenesis. Viruses. 2021;13(6):1124. Line 230.
2) Hu J, Liu K. Complete and Incomplete Hepatitis B Virus Particles: Formation, Function, and Application. Viruses. 2017 Mar 21;9(3):56. To support our manuscript. Line 236.
3) Prange R. Hepatitis B virus movement through the hepatocyte: An update. Biol Cell. 2022 Dec;114(12):325-348. Line 238.
References:
Ballabio, A. and J. S. Bonifacino (2020). "Lysosomes as dynamic regulators of cell and organismal homeostasis." Nat Rev Mol Cell Biol 21(2): 101-118.
Shirai, R., M. Cho, M. Isogai, S. Fukatsu, M. Okabe, M. Okawa, Y. Miyamoto, T. Torii and J. Yamauchi (2023). "FTD/ALS Type 7-Associated Thr104Asn Mutation of CHMP2B Blunts Neuronal Process Elongation, and Is Recovered by Knockdown of Arf4, the Golgi Stress Regulator." Neurol Int 15(3): 980-993.
Ugbode, C. and R. J. H. West (2021). "Lessons learned from CHMP2B, implications for frontotemporal dementia and amyotrophic lateral sclerosis." Neurobiol Dis 147: 105144.

Round 2
Reviewer 1 Report
Comments and Suggestions for Authors
While the authors amended most of the prior problems, I am still a bit worried about the legibility of fig 2 and table 3.
Author Response
Comments and Suggestions for Authors:While the authors amended most of the prior problems, I am still a bit worried about the legibility of fig 2 and table 3.
Answer: Thank you once again for your valuable feedback! We merged these two sections into the revised Figure 2. Given the complexity of ESCRT-mediated regulation of HBV, which spans multiple stages of the viral life cycle and exhibits variability arising from differences in cell lines, experimental approaches, and interventions, the original Table 3 has been moved to the Supplementary Materials for the readers who are interested in details. The updated Figure 2 now focuses on streamlining the HBV life cycle, specifically highlighting the impact of siRNA knockdown and dominant-negative (DN) plasmid overexpression in Huh7 cells on the secretion of distinct HBV particles (virions, capsids, and subviral particles).
We added a new statement in Lines 255-261:“The regulation of the HBV life cycle by the ESCRT machinery is highly complex. As illustrated in Figure 2, a concise summary is provided of the effects observed in the Huh7 cell line following siRNA-mediated knockdown of ESCRT components or overexpression of dominant-negative ESCRT mutants on the secretion of HBV virions, capsids, and subviral particles (SVPs). This experimental approach highlights the critical role of ESCRT in distinct stages of HBV assembly and egress, particularly in modulating the trafficking and release of viral and subviral components.”
The new Figure:
Figure 2. Schematic of ESCRT-mediated modulation of the HBV particle secretion. HBV virions bind and enter host cells, followed by the conversion of relaxed circular DNA (rcDNA) into covalently closed circular DNA (cccDNA) within the nucleus. The cccDNA serves as a transcriptional template to synthesize viral RNAs (depicted as wavy lines), including: PreC mRNA, pregenomic RNA (pgRNA), PreS1 mRNA, PreS2/S mRNA, and X mRNA. These transcripts encode distinct viral proteins: PreC mRNA for HBeAg; pgRNA for the HBcAg and polymerase; PreS1 mRNA for the Large HBsAg; PreS2/S mRNA for the Medium and Small HBsAg. Reverse transcription occurs within newly assembled capsids, generating rcDNA-containing nucleocapsids. Mature HBV virions (enveloped particles containing rcDNA), subviral particles (SVPs; spherical and filamentous forms exclusively containing HBsAg), and capsids (with or without rcDNA) are subsequently secreted. Notably, experimental inhibition of specific ESCRT components (via siRNA knockdown or dominant-negative mutants) disrupts the secretion of these viral and subviral structures are summarized. The figure was created in Biorender.
